Reproducing country-wide COVID-19 dynamics can require the usage of a set of SIR systems

http://orcid.org/0000-0001-7904-1881 Postnikov Eugene B. postnikov@kursksu.ru
Kursk State University , Kursk , Russia
Aly Sharif
Electronic publication date: 2021 Jan 7
Publication date: 2021
Volume: 9
Electronic Location ID: e10679
Received 2020 May 30; Accepted 2020 Dec 9
Copyright: © 2021 Postnikov
Copyright year: 2021
Copyright holder: Postnikov
License: This is an open access article distributed under the terms of the Creative Commons Attribution License, which permits unrestricted use, distribution, reproduction and adaptation in any medium and for any purpose provided that it is properly attributed. For attribution, the original author(s), title, publication source (PeerJ) and either DOI or URL of the article must be cited.
License URL: https://creativecommons.org/licenses/by/4.0/

Keywords: COVID-19, SIR model, Multilogistic regression

Funding: The author received no funding for this work.

==============================
This work shows that simple compartmental epidemiological models may not reproduce actually reported country-wide statistics since the latter reflects the cumulative amount of infected persons, which in fact is a sum of outbreaks within different patched. It the same time, the multilogistic decomposition of such epidemiological curves reveals components, which are quite close to the solutions of the SIR model in logistic approximations characterised by different sets of parameters including time shifts. This line of reasoning is confirmed by processing data for Spain and Russia in details and, additionally, is illustrated for several other countries.

Introduction

The COVID-19 pandemic is recently one of the most urgent problems and challenges of public health, which affects almost all countries of the world; see Sohrabi et al. (2020), Wang et al. (2020), He, Deng & Li (2020), Gates (2020) and Bedford et al. (2020) for a general review. Respectively, this situation states the problem of mathematical modelling of the local and global outbreaks aimed at understanding the epidemic process, revealing factors affecting the course of the outbreak, the effectiveness of applied measures and provided recommendations, which should prevent possible new waves of this disease (Cobey, 2020).

One of the natural approaches to such modelling is the usage of different variants of compartmental models (Brauer, Van den Driesche & Wu, 2008), which subdivide the full populations into groups of susceptible S, infected I and recovered/removed R persons (SIR model), which can also be supplied with the group of exposed persons (E), and even with more detailed subdivision. However, it should be pointed out that the expended SEIR model have some practical weak points, when one address the country-wide data. First of all, its solution can only be reduced to a 2-dimensional system (Weinstein et al., 2020), in contrast to the SIR model, which is has an exact 1-dimensional representation. This results in the necessity of fitting a ‘hidden’ variable E and additional kinetic parameter, which an appropriate change of variables for the SIR model puts its solution in correspondence to the 1-dimensional data reported. At the second, both models are compartmental and, in the strict sense, requires consideration of a well-mixed population. However, the compartmental models are still applicable when an outbreak starts practically synchronous over a wide region. But in this case, the summation of inputs from different regions makes the delays realised as the two-step transition S → E → I sufficiently less definite than in the case of small well-localised populations, and, in fact, it can be taken into account by using an appropriate effective kinetic coefficient considering the simple S → I transition.

A number of recent studies (Maier & Brockmann, 2020; Fanelli & Piazza, 2020; Postnikov, 2020; Linka et al., 2020; Carcione et al., 2020) demonstrated a reasonable reproduction of the dynamics of the actually reported cases in a variety of countries on the time intervals from the growing stage of the outbreak until the vicinity of its peak, and Dehning et al. (2020) analysed the well-developed outbreak with parameter switching of the SIR model. Even for the case of the simplest SIR model chosen for the reasons mentioned above, it has been shown that: first, the lack of detailed data on intra-country distributions, movements and contacts in the country-wide reports, which give a strong mean-field picture, makes desirable minimisation of the hidden parameters; second and important, the SIR model provides an opportunity for the direct data-driven approach.

Postnikov (2020) showed such a model defined by the system of ordinary differential equations (1) dSdt=−kSI,

(2) dIdt=kSI−τ−1I,

(3) dRdt=τ−1I,

where k and τ are the transmission and recovery rates, can be sequentially reduced to one Verhulst’s logistic equation with respect to the cumulative number of infected persons (4) dΣIdt=rΣI(1−ΣIK),

where r is the growth rate, which is directly connected with the basic reproduction number R0 via parameters of the differential Eqs. (1)–(3), see the explicit formulae below, in the section “Materials and Methods”. The parameter K is the saturation value. It depends on both R0 and the total number of potentially susceptible people, see “Materials and Methods” again; the problem of its proper determination, which can be connected to either spatially or socially separated subpopulations, that is there is a difficulty in its estimation a priori. Moreover, the cumulative mean-field epidemic curve can be affected by mixing such subpopulations that is the main focus of the present study.

At the same time, it should be kept in mind that the logistic function, which is the analytic solution to Eq. (4), is also an approximate solution to the SIR-system of ordinary differential Eqs. (1)–(3) derived on a basis of existing invariants of this ODE system and the assumption that maxI(t) ≪ K fulfills in practice; see Postnikov (2020) for the step-by-step derivation. The analysis carried out for different countries till the first decade of April confirmed quantitative adequacy of this approach for different countries among which were Italy, France, Germany, Spain.

However, new data registered to the decaying stage of the outbreak exhibit drastic deviations from such a logistic curve. The same behaviour is noted for other compartmental models that induced different attempts to their more complicated modification; for example, Oliveira (2020), Mamon (2020) and Schaback (2020).

Note that the preliminary analysis evaluated for the case of Italy (Vannucci & Vannucci, 2020) demonstrated that reproducibility could be restored if to consider a combination of two logistic curves corresponding to two outbreak waves. The similar combination was applied earlier to the long-time modelling of tuberculosis (Lavrova et al., 2017) based on a superposition of SIR models with different (switched with time) control parameters. Note also that a similar behaviour of data curves is already known in the theory of markets’ evolution and technological changes known as the logistic substitution model (Meyer, Yung & Ausubel, 1999) and the respective software Loglet Lab 4 was developed to carry out such analysis. It is available online at https://logletlab.com and supplied with a variety of examples of such decomposition and details of the numerical algorithms.

Thus, the main goal of this work is to explore if it is possible to decompose the data reported for COVID-19 outbreaks at the country-wide scale into a set of curves, each of which satisfies the SIR-Verhulst model. In other words, to explore if it is possible to consider the dynamics of such data as a superposition of outbreaks satisfying the simple compartmental model but operating with different subpopulations and parameters and, respectively, to acquire a tool for revealing a number of such individual characteristic modes from unique general data.

Materials and Methods

The project ‘Our world in Data’ aggregating the data related to COVID-19 pandemics from different sources (Roser et al., 2020) was used as the data source. These data were downloaded in an automatic regime with a home-written function, which belongs to the developed tool for MATLAB/GNU Octave made available in the GitHub repository (https://github.com/postnicov/owid_loglet_interface) and their subset containing the cumulative number of infected persons for a chosen country (total cases) was processed to form a text file suitable for uploading to the online software Loglet Lab 4 and carrying out the multilogistic decomposition. Comments on the particular issues of the programme realisation, processed data structure and the workflow can be found also in the mentioned repository, the full set of the data and the processing code for countries considered in this work is placed there in the ‘examples’ directory under the name ‘10countries_data_August’.

The results of this decomposition report the components of decomposition. Each such component is the logistic curve satisfying the SIR-Verhulst Eq. (4) and has the form (5) ΣI(t)=K1+e−r(t−tm),

where tm defines the time moment when the cumulative number of infected persons reaches half of the saturated values. It should be pointed out that the growth and saturation parameters r and K are directly connected with the parameters of the SIR model (1)–(3) as r = (R0 − 1)τ−1 and K=R02(2(R0−1)N)−1, where R0 = kτ is the basic reproductive number and N = S + I + R = const, see Postnikov (2020) for details.

An additional output is the so-called Fisher–Pry plot (Fisher & Pry, 1971) for each component. Its utility if based on its construction as the plot of the logarithm of the cumulative registered cases divided by the expected maximum as a function of time. In the case of the solution (5) this representation gives (6) lnΣI/K1−ΣI/K=−r(t−tm),

that is one can easily check if the data already satisfy the logistic curve looking do they follow a straight line or not. If linearity of the Fisher–Pry plot fulfils, this means that the SIR model provides a sufficient accuracy to reproduce the basic features of the analysed data, if not—one need to change the model.

In addition, the average relative absolute deviation (AAD), AAD=100%⟨ΣIrec−ΣIcalcΣIrec⟩

is used in this work as a quantitative criterion. Here σIrec and σIcalc are the actually registered and the model-based calculated values of the cumulative number of infected persons, respectively. The average value is taken over the whole time interval covering values ΣIrec>0.05⋅max(ΣIrec).

The particular considerations of the cases of Spain and Russia use also more geographically detailed data, which were obtained from from the web-site of the National Epidemiological Center of Spain (Centro Nacional de Epidemiología (CNE), 2020) and accumulated by the Federal Service for Surveillance on Consumer Rights Protection and Human Wellbeing accessed via Yandex’s (2020) web service DataLens, respectively. The data and the programme code, which carried out their processing, can be be found in the GitHub repository mentioned above, too. In particular, the illustrated worklow’s description for the detailed analysis of data for two cumulative data from two regions of Spain is placed there in the zip-archive named ‘regionsSpain’ as well as the data and the source code used for their processing.

Statement of the Particular Problem

As two principal illustrative examples, which illustrate the failure of reproducing the long-term country-wide dynamics of COVID-19 outbreaks, the cases of two countries, Spain and Russia, can be considered. Postnikov (2020) showed that it satisfied the Verhulst-SIR model (4) with high accuracy for the state-of-the-art up to the first decade of April. However, the actual longevity and cumulative effects of outbreaks exceed the predictions made using the data available for the early stages of the process.

Figure 1 demonstrates the data downloaded from ‘Our World in Data’, uploaded for processing into Loglet Lab and post-processed again with the Octave script (see “Materials and Methods”). Here the unique logistic curve was applied for the cumulative number of infected persons (total cases), that is the ‘composite plot’ is automatically composed of one component.

Figure 1 The total cases of persons infected by COVID-19 (A & B) according to ‘Our World in Data’ for Spain (from March, 2 to July, 1) and Russia on August, 11 (circles) and their logistic fit (solid curves) as well as growth rates (C & D) and Fisher-Pry plots (E & F) for these data and their fitting.

One can see that the single logistic curve fits the total cases only qualitatively on average, see Fig. 1A for Spain, AAD = 6.66% and Fig. 1B for Russia, AAD = 8.19%. The quantitative details of approximation do not accurately reproduce the dynamics, black circles go under and over the red curve within relatively long intervals as well as there are significant displacements of recorded and model curves on the bell plots, see Figs. 1C and 1D, respectively. In addition, this inconsistency is most obvious looking at the Fisher–Pry plots (Figs. 1E and 1F): the markers do not follow straight lines as it should be from (6) if the recorded data with a single defined K satisfy the logistic growth law. An attempt to improve the correspondence superposing maxima manually do not really improve the situation since there will emerge ‘long tails’ of recorded data drastically overcoming fast-decaying model rate curve and underestimating the curves of total cases.

These facts argue in favour of the idea that the data accumulated over the whole country may in fact consist of different maxima and different growth rates for different regions, which start to be significant after the middle of April (before this date there was a quite accurate correspondence between he data and the SIR-Verhulst model, see Postnikov (2020)). This hypothesis will be tested considering both multilogistic decomposition and the outbreak’s dynamics in spatially separated regions within each country.

Results

As the first example, let us consider the case of Spain since the registered and the model curves in Fig. 1C visually differ mainly by the location of their maxima. Figures 2A and 2C represents the examples of regional statistics for two separated regions of Spain, Madrid and Catalonia. One can see that before the first decade of April both curves had the similar shape, similar location of their maxima and differ in the magnitude only. Therefore, such a synchronisation leads to a possibility to represent the summed data as a single curve with some unique set of effective parameters of the SIR-Verhust model. The same is true for the rest of the regions too and, as a results, one system (1)–(3) reduced to one Eq. (4) with one logistic functions as the solution is completely enough as it has been already demonstrated earlier.

Figure 2 The plots demonstrating daily registered incidences of COVID-19 (A) and their cumulative sums (C) in two valuable spatially separated regions of Spain and the decomposition of the registered data curve as well as the data decomposition into two logistic curves (D) with the growth rates shown in (B).

On the contrary, the data for Madrid demonstrates the secondary outbreak between the first decades of April and May, which can not be predicted directly from the data recorded earlier. Moreover, since the SIR system (1)–(3) has an unimodal solution for I(t), it can be hypothesised that the disease spread over a group of susceptible persons, which differ from the initial one.

Therefore, it is natural idea to carry out more detailed analysis, which includes consideration of an additional independent SIR system. As follows from the reducibility of the SIR ODE system to one Verhulst’s equation, this means that it is required to consider two logistic curves whose sum should fit the data not fitted by one curve only. This decomposition in practice can be easily carried out by means of Loglet Lab software switching in the mode of several logistic curves (the detailed workflow of this procedure is described within the supplementary GitHub-deposited data, see “Materials and Methods” section). Applying this procedure to the sum of regional curves (it is shown as black circles in Fig. 2B), it revealed that enlarging number of possible curves to two instead of one results in a sufficinently accurate reproducing the analysed data: compare the upper curves in Figs. 2B and 2D, quantitatively AAD = 0.86% for the two-component model. These logistic components are shown in Fig. 2D too, and their derivatives, which have a meaning of the daily registered cases for each of these components are depicted in Fig. 2C. Considering the latter, it should be pointed out that the multicomponent decomposition algorithm does not distinguish the synchronised curves but is reveals outbreaks, which are are separated in time. But in this case the method reveals the principal features of temporal dynamics: one can see comparing maxima in Fig. 2C and curves in Fig. 2A that the first revealed component detected the common maximum of outbreaks in both Madrid and Catalonia, and the second—the summer maximum seen in the epidemic curve for Madrid.

Therefore, it is posible to apply this method to data for the whole Spain. The respective result is shown in Fig. 3. Figure 3 indicates that, in the case of Spain, the assumption of the existence of two logistic components gives sufficiently better reproducing the observed dynamics, AAD = 1.73%. One component corresponds to the more fast and strong outbreak with the peak on March, 27 and the second is slower with the peak on April, 18 that is agreed with the discussion of Fig. 2 above. Both components follow the solution of the Verhulst equation as seen from the respective plot in Fig. 3 and reasonable linearity of partial Fisher–Pry plots there. The last fact confirms the validity of logistic approximation for the both components considered as independent. The full set of parameters for both logistic functions are listed in Table 1.

Figure 3 The plots generated after the downloading data from ‘Our World in Data’ for Spain on July, 1, their processing with Loglet Lab and postprocessing.

Different components (B–D) revealed as comprising the total epidemic curve (A, cumulative number of revealed infected persons) are marked with different colours. The circles are recorded data and the lines are their reconstruction by the solutions of the SIR-based logistic Eq. (4).

Table 1 The parameters of the separated logistic components (5) for the case of Spain, where r (days−1) is the components growth rate, tm (days) is the time location of the inflection point for the logistic function, and K (persons) is the logistic curve’s maximum.

The first day of the time series is March, 2. R2 denotes the coefficient of determination.

Phase	K	tm	r	R2	
1	134,912	26.0	0.225	0.973	
2	108,805	47.4	0.087	0.973	

The next example, the case of Russia, which is sufficiently more spatially extended country than Spain, requires a larger number of logistic components. Figure 4 demonstrates this five-modes decomposition valid to the current state of still progressing epidemics in Russia, which reproduces the actual data with a pretty good accuracy, AAD = 0.334%. Each component curve corresponds to the logistic growth process with high accuracy too that is seen from all plot representations, for the components themselves, their growth rates, and the Fisher–Pry decompositions as well as from the high value of the coefficient of determination. Notably, that the growth coefficient values k given in Table 2 can be subdivided into two groups. At the first stage (from the middle of March till the middle of May), there are two subsequent component with large k, and then (since the middle of May up to date) three components with practically equal k again time-shifted. This is also demonstrably visible in the plot representing the Fisher–Pry transform. This division can be put in line with the governmental measures: the period March, 30–May, 11 was non-working nationwide, the public places lockdown and strict travel restrictions were applied. Notably, this period parenthetically coincides with the period of existence of the major part of the first two components in Fig. 4. Therefore, it can be concluded that the measures applied localised the number of potentially susceptible persons by several spheres of close contacts of infected persons and led to the fast reaching the saturated values for these two first logistic components. Certainly, the contacts were not broken completely and the outbreak continued to spread countrywide but with sufficiently slower growth rate even over the background of more mild restrictions. The three subsequent logistic components can be associated with such a spatial spread.

Figure 4 The plots generated after the downloading data from ‘Our World in Data’ for Russia on August, 11, their processing with Loglet Lab and postprocessing.

Different components (subplots (B–D) revealed as comprising the total epidemic curve (cumulative number of revealed infected persons, (A)) are marked with different colours. The circles are recorded data and the lines are their reconstruction by the solutions of the SIR-based logistic Eq. (4). Dashed lines correspond to the model-based projections.

Table 2 The parameters of the separated logistic components (5) for the case of Russia.

The units of r, tm, K are days, days−1, persons, respectively. The first day of the time series is March, 18. R2 denotes the coefficient of determination.

Phase	K	tm	r	R2	
1	65,744	35.0	0.183	0.979	
2	178,111	52.0	0.176	0.996	
3	379,133	79.0	0.085	0.999	
4	200,709	116	0.085	0.997	
5	192,037	149	0.080	0.997	

As above for Spain, the origin of such a behaviour for Russia can be explained exploring the geographically subdivision as it is shown in Fig. 5 using the regional data for Russia accumulated by the Federal Service for Surveillance on Consumer Rights Protection and Human Wellbeing accessed via Yandex’s (2020) web serve DataLens. Moscow as the capital, the most populated city, and the largest transport hub is considered separately along with the European (without Moscow) and Asian (including the borderline Ural region) parts of country. From epidemiological point of view, one can see that Fig. 5 demonstrates that such a subdivision provides comparable amount of infected persons. Thus, it is possible to put in correspondence features in Figs. 4 and 5. First of all, it is visible that the outbreak in Moscow is the leading one during the first stage of the time course. In addition, the respective curve is bimodal. Therefore, the April’s peak can be associated with the primary introduced virus spreaders (Moscow is a huge transport hub); this peak’s behaviour also corresponds to the model results provided in Postnikov (2020), where only the data up to the first decade of April were used. Further, the spreaders simultaneously distributed the virus over Moscow (secondary peak of the red curve in Fig. 5) and over the country with a prevalence to regions closer to Moscow (the European part, blue curve in in Fig. 5). Note, however, that this region consists of multiple administrative units and, whence, the blue blue curve has not so accurate bell-like shape as one city, Moscow. The third (green) curve corresponds to more distant from Moscow and more weakly connected Ural, Siberian, and Far East part of Russia, and it demonstrates the larger time delay in the outbreaks’ development. At the same time, more-or-less uniform restrictive measures ‘synchronize’ the parameters of SIR model even for distinct weakly or non-connected locations whereby a sum of almost independent data provides an opportunity to use the mean-field description resulting in the accurate sequence of logistic representations for the countrywide data shown in Fig. 4. At the same time, it should be kept in mind that the parameters K from Table 2 are not directly associated with particular geographic regions within such a multiple-SIR but mean-field approach, they represent a characteristic mixture of potentially susceptible groups distributed between different regions but simultaneously actual for the particular stage of the epidemic spread.

Figure 5 The plots demonstrating daily registered incidences of COVID-19 in Russia subdivided accordingly to principal socio-geographical division.

Finally, Fig. 6 provides demonstrations additional to the considered above and covers the most affected countries during the spring–summer outbreak. One can see (not only visually but also basing on the values of the average relative absolute deviations) that the proposed decomposition reproduces the specific features of the cumulative number of cases with high accuracy and the multilogistic representation (i.e. superposition of several SIR systems associated with different sub-communities) is a widespread phenomenon. A single curve corresponds in the cases of Columbia (Fig. 6C) and India (Fig. 6D) to the situation, when the outbreak has not reach the peak yet (or, equivalently, the middle level with respect to the saturation estimated from the exiting data fitting). However, it should be pointed out that the forecast of the maximum total number of infected in such a case has a high level of uncertainty, see numerical tests in Alberti & Faranda (2020), Postnikov (2020) and an additional analysis of sensibility of sigmoid, including logistic, curves to perturbations in Faranda et al. (2020). This caveat is possibly true also for the second stage of the two-component representation of data for Peru (Fig. 6F).

Figure 6 The plots (A–H) demonstrating the cumulative numbers of cases (circles), the respective solutions of logistic and multilogistic models (red solid line, the dashed part corresponds to the projection based on the simple continuation with the present state parameters) and their components (if any; dots denotes the sorted data and curves are the simulated continuous solutions) for eight countries among top-10 most affected ones.

The parameters for all models/components are given in supplementary material on the GitHub project (see “Materials and Methods”) as well as the data and code for their processing.

Among the rest countries considered, where the epidemic curve looks overcoming the middle point, three of six demonstrate the multilogistic character: two components for Brazil (Fig. 6A) and Mexico (Fig. 6E), and three for the USA (Fig. 6H). The latter is more spatially spread and, additionally, social turbulence of last months may influence the parameters of the third component there.

Discussion

The examples given above reveals that the simple logistic regression with a unique fitting function corresponding to the solution of the SIR model can not give a fully adequate picture of the outbreak in its full time course even if it was accurately applicable during almost complete initial growth phase, see Postnikov (2020) for the same countries. The main reasons for this conclusion are the following: (i) the epidemics spreads over spatially distributed countries while the reported data curves gives summed values, that is and the validity level of a mean-field model depends on a possible contrast in the course of developed epidemics in different part of the country (more ’compact’ countries are more uniform in the response that more ‘elongated’ as seem from the comparison of Spain and Russia above and other examples discussed below); (ii) effects of counter-epidemic measures can change the outbreak size and prolong its continuity that is reflected in a less value of the parameter k in Eqs. (1) and (2), or, equivalently Eq. (4) that is also visible in Tables 1 and 2: the values of k for the second Spanish component and the third-fifth Russian components are practically the same (the governmental restrictive measures were more or less similar in the both countries).

Note also that the proposed method indicates time-shifted maxima, while synchronous outbreaks are identified as a single one. This situation can emerge when initial conditions for the outbreak’s start fulfil in geographically different sub-communities (a widespread initial introduction of a new type of infections agent is a plausible reason for such a behaviour). In addition, this observation explains why compartment models developed for well-mixed populations may reproduce epidemics in spatially-extended regions.

Thus, these complications do not mean that the considered disease does not always follow the classic well-established SIR model of mathematical epidemiology (and its logistic-form) and the mean-field consideration, which addresses the country-wide data. The analysis carried out in the present work revealed that a more realistic data regression requires the decomposition over multiple logistic curves. Each of such curve corresponds to its own SIR system with its own set of parameters. Among these parameters, the possible maximal number of susceptible person within weakly connected communities (on even approximated as uncoupled ones for simplicity of fitting) is one of the most important. Such situation can have an interpretation in terms of metapopulation approach (Colizza & Vespignani, 2008; Bichara et al., 2015; Ball et al., 2015; Chowell et al., 2016). In fact, the country-wide statistics unifies data from different regions and even from different social strata within each region. Due to differences and different connectivity between such sub-populations, the local outbreaks can be shifted in time and intensity. This problem especially arise under conditions of different measures of travel restrictions and social distancing applied practically in all countries during COVID-19 pandemics.

Conclusion

The main message of this work is aimed at attracting attention to the importance of multimodal consideration of epidemic curves given at the macroscopic (country-wide) level and developing a method, which allows revealing time intervals of qualitative changes in dynamics as well as their quantitative estimations.

The complicated non-standard shape of epidemic curves not necessarily originates from some special features of the current pandemics providing a challenge in developing new very complicated multicomponent models, which contain a large number of parameters inaccessible directly. In fact, when one addresses the global mean-field data on the ‘macroscopic’ scale of a country, a set of simple SIR model is plausible. In its turn, the number and temporal localisation of the revealed principal modes provide perspectives for building more detailed ‘microscopic’ models, which should take into account spatial, social, etc. features, that is to reflect phenomena originated from the metapopulational dynamics. Note that the summation of individual growth curves attracted attention as an empiric approach from the classical problems of pure population dynamics of heterogeneous populations (Reed & Pearl, 1927) up to the mathematical problems of epidemic spread in metapopulations (Chowell et al., 2016). In this work, it is shown that the effective logistic curve-based approach could play a role of an accurate tool for studying the recent COVID-19 pandemics in a tight connexion with the classic SIR model approach. In contrast to the direct simulation of ODE’s including the case of parameter switching in a priori known time points (e.g. due to lockdown measures, see Dehning et al., 2020), the proposed method solves the inverse problem revealing actual parameters and inflection points originated from the actual data.

Finally, this opens certain perspectives for future investigations in modelling by analogy to other field of science, which use dynamical systems as a principal tool. As some examples, the dimension reduction to several principal modes is used for simplified modelling of turbulence (Sapsis & Majda, 2013), distributed elastic mechanical systems (Kerschen & Golinval, 2002), chemical processes (Qing et al., 2020), etc. Thus, the revealed possibility of describing the cumulative dynamics of the epidemics with a finite number of components calls for development of spatio-temporal models whose full-scale dynamics is reducible to the dynamics of several principal modes only reflected in the generalised data.

I thank Dr. Irina L. Zhirnova for the motivational discussions of the epidemic growth curves dynamics and composition.

Additional Information and Declarations

Competing Interests

Author Contributions

Data Availability

The author declares that they have no competing interests.

Eugene B. Postnikov conceived and designed the experiments, performed the experiments, analysed the data, prepared figures and/or tables, authored or reviewed drafts of the paper, and approved the final draft.

The following information was supplied regarding data availability:

Data is available at GitHub: https://github.com/postnicov/owid_loglet_interface.

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
