# Peer review of "Reproducing country-wide COVID-19 dynamics can require the usage of a set of SIR systems"

_PeerJ, doi:10.7717/peerj.10679_

## Round 0.1 · original submission · Major Revisions

Experts have evaluated your manuscript and identified key issues in its methods, I invite you to address their comments. The manuscript can benefit from some wordsmithing, but I am more concerned with the overall research goal, methods detail and discussion. Can you please address these comments:

1) A clear research goal statement is needed, are you proposing an alternate model fitting method (i.e. logistic curve fitting) instead of the differential equations? and why?

2) If so, why was the logistic curve your choice, as supposed to other options?

3) Your choice of logistic curve fitting in a posteriori modeling approach should be justified, i.e. are the SIR models not good enough? (all models are going to have errors, but fig 1 showing the observed and SIR fitted are pretty good already. Also to make your point you should show the observed, logistic and SIR all in one plot in Fig 2).

4) Can the logistic curve fitting predict or make forecasts in an apriori approach? SIR models can incorporate quite a variety of biological details, such as vector transmission, vaccination etc. How can the loglet analysis do such things, or is it only going to fit curves post-hoc? Case in point, can you introduce a vaccine and model the stability state of the model to educate stakeholders if the outbreak will be under control, and eradicated?

5) Its not clear how can one estimate R_0 from a logistic curve, please show how?

6) I am very concerned with the lack of details on your method of identifying the number of curves fit in each analysis, for example, how did you decide that the Spanish outbreak needed 2 curves while the Russian needed 4? More problematic is how did you arrive at each equation (curve's) terms? What is the algorithm you used, this needs to be described and some measure of goodness of fit is needed to demonstrate at what point the estimation converged?

7) The discussion is lacking tremendously. A serious discussion of the limitations of this methods (hints already listed above and in reviewer comments) is needed. Also how can this be streamlined to help epidemiologists fighting COVID-19.

I hope these comments help formulate a more complete manuscript for the next submission, otherwise, the concept and effort you put in here may be jeopardized by the presentation and lack of details, when in fact it could at least offer up a new method that may support the classical modeling efforts to dissect outbreaks.

Reviewer 1 ·

Basic reporting

(1) Line 25, 'One of the natural first approach to such...' should be 'One of the natural first approaches to ....'
(2) The meaning and derivation of R0 in this model should be discussed.
(3) Conclusion can be strengthened by adding some future directions and open problems.

Experimental design

(4) Line 65, why this can be easily checked to see if the data satisfies the logistic curve is not straigtforward.
(5) Regarding the system, what would happen if a feedback introduced like Global stability of disease-free equilibria in a two-group SI model with feedback control? I suppose the numerical results will be better especially for Spain as there is a clear feedback factor.
(6) In the tables 1 & 2, the meaning of these parameters should be recalled directly in the tables themselves.

Validity of the findings

(7) The work is based on numerical simulations. However, it should be mentioned that alalytical exact solutions can be derived. The solutions can be worked out via the Lie algebra method as shown in A Lie algebra approach to susceptible-infected-susceptible epidemics, Analytical solution for an in-host viral infection model with time-inhomogeneous rates. I understand that an exact solution is beyond the scope of this paper but a remark would be useful.
(8) In Figure 1, how do you fit these data? Are the curves polynomials?
(9) The resuls for Russia are interesting. There is inconsistency between ascending part of the outbreak for circles, which exhibit a transition in dynamics within the week from April 16 and April 30. More explanations are appreciated.

·

Basic reporting

1) The overall language of this paper in terms of grammar, sentence structuring and typos should be improved to ensure that an international audience can clearly understand your text. Some examples where the language could be improved include:

• Line-18: edit (challenges of public health)
• Line-22: edit (the effectiveness of applying)
• Line-25: edit (first approaches to such modeling)
• Line-27: edit (which can also be supplied)
• Line-28: edit (exposed persons (E))
• Lines 28-31: kindly restructure the sentence.
• Line-31: remove (at the)
• Line-45: edit (reproducibility could be restored)
• Line-49: edit (known as the logistic substitution model)
• Line-59: restructure sentence (As the source of data, there was chosen the repository, which belongs to the projects “Our world in Data”)
• Line-59: edit (This data was downloaded)
• Line-61: edit (growth and saturation parameters)
• Line-64: edit (component transforming)
• Line-68: edit (Figure-1: …..persons infected by COVID)
• Line-69: edit (accuracy for state of the art)
• Line-70: edit (effects of outbreaks exceed)
• Line-76: edit (do not reproduce the dynamics accurately;)
• Line-78: edit (with the root mean)
• Line-79: edit (the solid red line)
• Line-94: edit (demonstrates a more)

2) Data source is shared, and figures meet the basic requirements.
3) I feel that the paper in its current form lacks context. There is not much relevant background and adequate number of references are lacking.

Experimental design

It can be interpreted that in past author has already published another paper that contradicts the findings from the current one and this is mentioned in his notes. The author has loosely tried to justify this paper by highlighting data limitation during previous research.

When you are trying to contradict your own research in such a short period of time, then you need to really put a good effort to justify that. The current paper is short and seems to be done in a rush. It feels that only the model has been re-run on a larger and more updated dataset and conclusion is made based on the results.

With COVID-19 data still being updated – I will highly recommend the author to verify and tabulate his findings by running the results on top 10 countries that are leading in total cases/ deaths.

Validity of the findings

Although the results may seem to make sense, but the level of effort put on this paper doesn’t give enough confidence to agree with it.

Additional comments

• The paper in its current form needs a major re-structuring with efforts on context, language and justification.

• Best wishes – hope you come back with a quality contribution.

---

## Round 0.2 · Major Revisions

Please address reviewer 3's comments making the necessary justification and modification of the proposed multilogistic decomposition, rather than comparison to simple logistic. In your contrasts with the compartmental models please switch to SEIR which is what is used now in the literature (as supposed to SIR) and update Fig 2. In comparing the logistic method to a simple SIR model without the effect of control measures (modeled by one of many methods including introducing a factor for change in interaction with infectious individuals, infectious environment, and many other modifications that can be introduced to a compartmental model) it is no surprise that the model will not predict the outbreak post peak. In other words, we don’t know if an SEIR model will predict just as well or better than the proposed method, if such variables I named were introduced.

Finally, you make the case from Fig 2 that the Spanish cities post-peak data shows poor prediction. However, you don’t go back and show the multilogistic equations predictions to these two cities, please add that.

Reviewer 1 ·

Basic reporting

n/a

Experimental design

n/a

Validity of the findings

n/a

Additional comments

The author has addressed most of my questions. Therefore, I am glad to recommend it for publication in PeerJ.

·

Basic reporting

no comment

Experimental design

no comment

Validity of the findings

no comment

Additional comments

In the manuscript, the author is highlighting the importance of using multi-logistic decomposition of countrywide COVID-19 cases to demonstrate its utility in delineating outbreak time phases and spatially distinct outbreaks. The following are my key comments and concerns regarding the study.
1. Conceptual advancement: I fail to agree on why in the first place the author thought that a simple SIR model and its analytical solution in terms of the Verhulst logistic equation can simply fit well on countrywide data. SIR models inherently assume the population is homogeneous (i.e there are not metapopulations). While a country (however small) will certainly abide by this assumption. Hence, the author shows through its analytical results that multi-logistic decomposition is a better fitting option is not a new finding. I would suggest the author change the narrative of the study. Rather than showing that multi-logistic decomposition is a better option than simple logistic decomposition (obvious result), I would rather ask the author to explore if multi-logistic decomposition can help decipher key epidemiological events/characteristics in a countrywide outbreak. The author briefly describes how the multi-logistic decomposition of Russian COVID data can help understand the outbreak in various cities and regions. Hence, understanding the number of phases and determining how many phases (sub SIR models) could be the key parameter to understand the epidemiology of COVID19 in a country. Furthermore, I also do not understand why the author is fitting SIR compartmental model rather than the SEIR model which is what most of the COVID19 models are based on.
2. Writing style: The author needs to majorly restructure its manuscript. I would recommend differentiating methods and results and bringing in all introduction related writing into the first section. Please see the attached pdf file with comments on it for more details. Furthermore, I also understand the manuscript has been written from an analytical perspective, but the author of PeerJ will be mostly epidemiologists and biologists. Hence, I would recommend including details related to all data transformations with explanations in the methods section. For example, the Fisher-Pry plot needs a detailed explanation of its utility and interpretation. Other tiny things such as figure subplots numbering and citing the correct figure subplot would significantly improve the manuscript reading experience.
3. Log let software: Looks like most of the model fitting is done using Log let the software, but details are missing to reproduce the results presented in the manuscript. Please provide all step-by-step details about analyses conducted in the manuscript (either in the main text or in supplement).

---

## Round 0.3 · accepted · Accept

Thank you for addressing the reviewers' and my comments. I am pleased to inform you that your manuscript is accepted and best wishes in your research.

·

Basic reporting

The author has put up a good effort on improving the manuscript.

Experimental design

Restructured paper in its current form along with more citations and detailed discussion looks good.

Validity of the findings

Although the research in itself in not novel but it definitely contributes with its content and case study for the academic audience.

Additional comments

Good work!

·

Basic reporting

The author has addressed all my comments and concerns.

Experimental design

N/A

Validity of the findings

N/A

Additional comments

The author has addressed all my comments and concerns.